# Impact of Inpatient Treatment for Obesity in Patients with Comorbid Psychiatric Disorders

**DOI:** 10.3390/bs15111562

**Published:** 2025-11-17

**Authors:** Marcia Cristina Almeida Magalhães Oliveira, Carina Marcia Magalhães Nepomuceno, Francielle Maria da Cruz Trindade, Carolina Chacra Carvalho e Marinho, Cristiano Gidi de Portela, Sérgio Oliveira Braga, Neidjane Sholl Pinheiro, Frederico Fidellis Barboza, José Lucas Sena da Silva, Natália Cristina de Oliveira

**Affiliations:** 1Life Sciences Department, Bahia State University, Salvador 41150-000, BA, Brazil; mcamoliveira@uneb.br; 2Obesity Hospital, Camaçari 42825-901, BA, Brazil; cari.magalhaes@gmail.com (C.M.M.N.); francielle_mc_trindade@hotmail.com (F.M.d.C.T.); carolina.marinho@hospitaldaobesidade.com.br (C.C.C.e.M.); drcristiano.hba@gmail.com (C.G.d.P.); sergio.braga@hospitaldaobesidade.com.br (S.O.B.); jane.sholl@hospitaldaobesidade.com.br (N.S.P.); frederico.barbosa@hospitaldaobesidade.com.br (F.F.B.); lucassena_cpm@yahoo.com.br (J.L.S.d.S.); 3Graduate Program in Health Promotion, Department of Medical Sciences, Adventist University of São Paulo, São Paulo 05858-001, SP, Brazil

**Keywords:** obesity, lifestyle, inpatients, depression, binge-eating disorder

## Abstract

Obesity is a global health problem causing millions of deaths from noncommunicable diseases. Individuals with obesity are also at increased risk for mental disorders, such as depression (DEP) and binge eating (BED). The aim of this study was to evaluate the effectiveness of an inpatient lifestyle-based intervention program for the treatment of obesity in persons with and without DEP and BED. This is a retrospective cohort study involving patients enrolled in a specialized inpatient hospital facility for the treatment of obesity. Participants underwent a long-term program composed of a low-calorie diet and intensive lifestyle changes. Data from 997 adult patients were included. Participants were divided into four groups: a depression group (DG), binge-eating disorder group (BG), depression and binge-eating disorder group (DBG), and a control group (CG). Anthropometric data were obtained at admission and discharge. Most participants were females, were sedentary, and were hospitalized for more than 3 months. No between-group differences were observed among DEP, BED, DEP + BED, and CG. Treatment duration was positively associated with reductions in weight and BMI in all patients. In conclusion, patients with DEP and BED with DEP + BED presented significant reductions in BMI and waist circumference, as well as the CG, and reduction in body weight was directly associated with the length of the intervention.

## 1. Introduction

Overweight and obesity worldwide have more than doubled among adults since the 1990s ([39]). Although the prevalence varies among countries, in the Americas, 67% of the population are living with excessive body weight ([39]). Obesity is characterized as a chronic, relapsing, multifactorial, heterogeneous disease, in which treatment and sustained weight loss remain among the main challenges in clinical practice ([23]).

The development of obesity is a complex process, involving genetics, environmental, cultural, and socioeconomic factors, besides eating behavior and lifestyle ([20]), leading to an imbalance of energy intake (determined by the mechanism of hunger, satiety, and hedonic eating) and expenditure (equivalent to resting energy expenditure, thermogenic effect of food, and exercise) ([39]).

In Brazil, since the 1970s, there has been a marked increase in overweight and obesity across all age groups and social strata, surpassing malnutrition ([34]). Recent studies indicate a 20% prevalence in obesity among Brazilian adults, with no statistically significant differences by sex, geographic region, or place of residence ([10]). A Brazilian study revealed that persons with obesity seem to be almost 40% more likely to present mental disorders ([24]).

The rise in obesity rates is a global health problem that in 2021 caused around 3.7 million deaths from noncommunicable diseases (NCDs) such as cardiovascular diseases, diabetes, cancers, chronic respiratory diseases, and others ([11]). Individuals with obesity are also at increased risk for mental disorders, and likewise, depression (DEP), one of the most common mental disorders, has been shown to elevate the risk of developing obesity, establishing a bidirectional relationship between the two comorbidities ([6]; [22]).

Internalization of weight stereotypes may influence the link between obesity and mental health outcomes, as the stigma is associated with negative health consequences ([26]). Stigmatizing weight experiences may contribute to depressive symptoms and lower levels of quality of life, as well as engagement in negative health behaviors, such as disordered eating and low physical activity levels ([33]).

Patients with depressive symptoms tend to present significant weight changes ([2]). The presence of symptoms such as psychomotor retardation, fatigue or loss of energy, loss of interest or pleasure in activities, among others, which can contribute to weight gain or a sedentary life. The severity of obesity, i.e., body mass index, also exerts an influence, increasing the strength of the obesity-depression relationship, especially in patients with grade III obesity ([42]; [7]).

Another condition commonly associated with obesity is binge eating disorder (BED), characterized by regular binge eating episodes in which individuals ingest large amounts of food and experience loss of control during the overeating episode ([2]). BED is the most prevalent eating disorder among adults and is strongly associated with obesity and with increased risk for psychiatric and other medical comorbidities ([12]; [14]). Remission from binge eating is associated with weight loss ([15]), which also results in numerous other long-term health benefits, and can be obtained from lifestyle intervention programs ([14]).

Lifestyle modification is the standard care in obesity treatment ([40]). It is also recommended to mitigate depressive symptoms ([36]). Nevertheless, the presence of mental disorders in patients with obesity may worsen adherence to lifestyle interventions, hindering weight loss and reducing patients’ motivation ([17]). In this sense, comprehensive inpatient treatments are usually superior to outpatient interventions regarding weight loss and treatment of obesity-related comorbidities ([29]). However, to the best of our knowledge, no study so far has investigated the impact of an inpatient intervention aimed at reducing body weight in persons with obesity who also struggle with DEP and/or BED.

Thus, the aim of this study was to evaluate the impact of an inpatient lifestyle-based intervention program for the treatment of obesity in persons with and without DEP and/or BED. Understanding how lifestyle modifications impact patients with obesity and mental disorders is crucial for providing them with efficient treatment options.

## 2. Materials and Methods

This retrospective cohort study involved adult patients enrolled in a specialized multiprofessional inpatient hospital program for the treatment of severe obesity. Data were collected between 2016 and 2022 after ethical approval.

All patient data included in this retrospective analysis were fully deidentified prior to use. Personal identifiers such as names, addresses, document numbers, contact information, and medical record numbers were removed by the data custodian before the dataset was accessed by the research team. Only coded, non-identifiable variables were used for statistical analyses. The anonymized dataset was stored on a secure, password-protected institutional server accessible only to the authorized research team members.

Ethical approval for this protocol (no. 65578822.1.0000.0057) included explicit requirements regarding anonymization and data confidentiality, in accordance with national regulations and the Declaration of Helsinki.

### 2.1. Participants

Data from 997 adult patients (aged ≥ 18 years) diagnosed with obesity class II or higher (body mass index, BMI ≥ 35 kg/m^2^) at baseline were included. Patients who remained in treatment for less than one month were excluded.

Diagnosis of depression (DEP) and binge-eating disorder (BED) was established at admission by a board-certified psychiatrist through a clinical interview based on DSM-5 diagnostic criteria ([1]). A comprehensive screening for all DSM-5 psychiatric disorders was not performed. Patients received monthly individual psychiatric follow-up during hospitalization.

Participants were categorized into four diagnostic groups according to the presence or absence of DEP and BED: depression group (DG); binge-eating disorder group (BG); depression and binge-eating disorder group (DBG); and control group (CG), with no diagnosis of DEP or BED.

### 2.2. Therapeutic Interventions

Participants underwent a multidisciplinary inpatient lifestyle-based treatment lasting from 1 to 6 months, according to individual needs and clinical evolution. The program followed the Brazilian Guidelines for the Management of Obesity ([3]) and emphasized long-term lifestyle modification through dietary management, physical activity, psychological support, and health education.

Patients received individualized low-calorie diets planned by registered dietitians. Menus were adapted to patient preferences, comorbidities, and treatment stage, and included weekly nutritional education sessions addressing portion control, mindful eating, and meal planning.

Daily supervised exercise sessions (at least 60–90 min/day) were conducted by physical education professionals. The protocol included aerobic training (walking, stationary cycling, water aerobics) and resistance exercises targeting large muscle groups, adjusted for each patient’s functional capacity. Stretching and relaxation sessions were also incorporated to promote adherence and recovery.

All participants engaged in weekly cognitive-behavioral therapy (CBT)-based sessions and individual psychological counseling focused on emotional regulation, stress management, and eating behavior. Those with DEP and/or BED received additional monitoring by the psychiatry team and, when necessary, pharmacological support as determined by the attending psychiatrist.

The multidisciplinary team (physicians, dietitians, psychologists, physiotherapists, and nurses) met weekly to discuss each patient’s progress and adapt treatment plans. Participants also attended educational workshops on sleep hygiene, physical activity, and relapse prevention.

### 2.3. Assessments

Demographic (age, gender) and lifestyle data were obtained from clinical records. Physical activity behavior was categorized as sedentary or non-sedentary according to self-reported regular engagement in ≥150 min/week of moderate-intensity exercise.

Anthropometric measurements were performed at admission and discharge by the same trained evaluator, following [38] ([38]) recommendations. Height was measured using a stadiometer (to the nearest 0.1 cm), and body weight using a calibrated digital scale (to the nearest 0.1 kg), with participants barefoot and wearing light clothing.

Circumferences were measured with an inextensible metric tape (graduated in millimeters) at standardized anatomical points as follows:Waist: 2 cm below the last rib;Hip: at the level of the greater trochanter;Calf: at the widest circumference of the dominant leg.

All measurements were taken in duplicate, with participants standing relaxed and feet shoulder-width apart, and the average of the two readings was recorded.

### 2.4. Statistical Analysis

Data were analyzed using the Statistical Package for the Social Sciences (SPSS) version 27.0 for Windows and expressed as n (%) or mean ± standard deviation. Demographic data were treated with descriptive statistics, and one-way ANOVA was employed to evaluate differences in continuous variables (age, time of hospitalization, % variation in weight, and % variation in BMI) among groups. Differences in the prevalence of categorical variables (gender and sedentary behavior) were assessed using Pearson’s chi-square test. The comparison of anthropometric data before and after the intervention was performed through two-way ANOVA for repeated measures.

The magnitude of effects was estimated using partial eta squared (η^2^_p_) as a measure of effect size. To account for potential confounding factors, an analysis of covariance (ANCOVA) was conducted, including age and gender as covariates, in order to verify whether the main effects of group and time on anthropometric outcomes remained significant after adjustment. As additional verification, ANCOVA models controlling for these covariates were also tested for each outcome. These adjustments did not substantially modify the results, indicating that the influence of age and gender on the main outcomes was minimal. Associations between the reductions in weight and BMI and the length of the treatment were evaluated using Pearson’s correlation coefficient. In all analyses, the significance level (α) was set at 5%.

## 3. Results

The sample of this study was composed of 997 adult patients with obesity, divided into 4 groups: patients diagnosed with depression (DG, n = 279), with binge-eating disorder (BG, n = 58), with both conditions (DBG, n = 38), and a group without DEP and/or BED (CG, n = 622). Most participants were females, sedentary (not achieving 150 min per week of physical activity), and hospitalized for more than 3 months (Table 1). Patients in DBG were younger than participants of the other groups, and the predominance of females was smaller in the CG.

An ANCOVA was performed to assess differences in BMI among the groups (DG, BG, DBG, and CG), controlling for age and gender. Results showed that the main effect of Group was not significant after adjustment, F(3, 902) = 0.575, *p* = 0.670, η^2^_p_ = 0.365. Neither age (*p* = 0.246) nor gender (*p* = 0.271) were significant covariates, and the Group × Gender interaction was not significant (*p* = 0.185). These findings indicate that group differences in BMI did not remain after controlling for demographic variables.

The multiprofessional hospitalization treatment for patients with obesity was efficient in reducing body mass index (BMI) and all other anthropometric parameters (Table 2). No between-group differences were detected among patients with depression, binge-eating disorder, and both conditions when compared to the control group. The effect of time showed a very large effect size (η^2^_p_ > 0.83 for all variables), indicating that more than 83% of the variance in the dependent variables was explained by changes over time.

The relative changes (%) in weight, BMI, and WC after the intervention are presented in Table 3. Groups exhibited similar changes in weight, BMI, and WC after the inpatient treatment.

The duration of the inpatient treatment was significantly associated with changes in weight (r = −0.401, *p* < 0.001) and BMI (r = −0.387, *p* < 0.001), but not with WC (r = −0.065, *p* = 0.625) in the total sample. This same trend was observed in the analysis separated by study group, which revealed significant weak or moderate associations between these variables in most cases, except for WC (Table 4).

Figure 1 and Figure 2 illustrate the relationships between inpatient duration and percentage changes in body weight and BMI, respectively, across diagnostic groups. In both analyses, longer hospitalization was generally associated with greater reductions in these variables.

In the scatter plot for weight change (Figure 1), the coefficients of determination (R^2^) ranged from 0.07 to 0.17, indicating small-to-moderate explanatory power. Similar trends were observed for BMI change (Figure 2), with R^2^ values of 0.166 for the CG, 0.141 for the DG, 0.065 for the BG, and 0.119 for the DBG.

These patterns indicate that, regardless of diagnostic group, longer inpatient treatment tended to be associated with greater reductions in body weight and BMI. The lower R^2^ observed in the binge-eating disorder (BG) group likely reflects greater interindividual variability in response rather than a weaker relationship between hospitalization length and anthropometric change.

## 4. Discussion

This study aimed at evaluating the effectiveness of an inpatient lifestyle-based intervention program for the treatment of obesity in persons with and without DEP and BED. Patients with one or both diagnoses presented equally significant improvements in the anthropometric variables evaluated when compared to the group without DEP and/or BED. Treatment duration was associated with both percentage weight change and BMI change, indicating that participants with longer hospitalization periods tended to experience greater reductions in body weight and BMI.

The female gender prevailed in the sample of this study. This finding is aligned with data from the National Health and Nutrition Examination Survey (NHANES), where researchers found this condition to be more prevalent in women than in men ([9]). Besides, women tend to be culturally more cautious about their health than men ([18]; [16]), which may also contribute to this finding.

The prevalence of sedentary behavior was over 80% in all study groups, in reinforcing the relationship between sedentary lifestyle and obesity, a well-documented fact, especially among adults ([13]). In this study, women showed a more sedentary lifestyle than men ([4]). Data from a survey in South American countries (including Brazil) revealed that about 40% of the adult population were sedentary (not achieving 150 min per week of physical activity) ([35]). The mean age of the participants was slightly different among groups, with DG presenting older and DBG younger participants. It is worth noticing that in Brazil, where the study was conducted, the greatest increase in the prevalence of obesity in recent years was in a very similar age group to that of the participants of this study (40–59 years old) ([8]).

The length of the intervention varied from 1 to 6 months, according to individual needs and desire to remain in the inpatient program. The main advantage of such a treatment modality is that the multidisciplinary team works in synergy with patients, providing them with daily specific and individualized health care in multiple areas of their expertise ([29]).

In contrast to previous research, depressive symptoms and a comorbid diagnosis of binge eating disorder were associated with less significant weight loss outcomes ([27]). However, the intervention implemented in the present study demonstrated efficacy in significantly reducing patients’ BMI. Notably, individuals diagnosed with DEP, BED, or DEP + BED exhibited comparable reductions in BMI compared to those in the group without these conditions. These findings suggest that a hospital-based intervention may be an effective treatment strategy for obesity, even in the context of co-occurring psychiatric disorders ([28]). A key feature of this intervention is the comprehensive mental health assessment provided to all participants by a multidisciplinary team, including both a psychologist and a psychiatrist. This integrated approach to addressing psychiatric comorbidities may contribute to increased adherence to the hospital treatment protocol and, consequently, to more favorable weight loss outcomes when compared to outpatient care.

There were no differences in the magnitude of the reduction in weight and BMI among groups. Patients reduced, on average, more than 20 kg of body weight, and more than 8.5 points in BMI. This was significantly associated with the duration of the inpatient treatment, regardless of the presence of DEP or BED. Longer hospitalization promoted greater reductions in both body weight and BMI. However, greater interindividual variability was observed among patients with BED, as reflected by the lower R^2^ values in both models. This variability may reflect the distinctive behavioral and neurobiological characteristics of binge-eating pathology, including higher impulsivity, emotional dysregulation, and challenges in maintaining adherence to structured dietary or physical activity regimens during treatment. Prior research indicates mixed findings: some studies report attenuated or more variable weight-loss trajectories among individuals with binge-eating disorder during weight-management programs, whereas meta-analytic data shows no overall difference compared with those without BED ([25]; [21]). These findings suggest that, while prolonged treatment may facilitate weight reduction, additional psychological and behavioral interventions targeting eating regulation and emotional coping are likely required to optimize outcomes in this subgroup.

It is noteworthy to observe that outpatient interventions are usually less effective. A study focused on lifestyle changes conducted for 12 weeks involving regular visits to a hospital and multidisciplinary education observed a mean reduction of 5.35 points in BMI ([19]). In the primary care setting, even with the aid of prescription medication for an average of 130 days, a retrospective study registered the mean loss of 1.2 kg of body weight.

At baseline, mean BMI in all 4 study groups was classified as obesity class III (>40) ([37]). After the inpatient intervention, DG, BG, and CG exhibited mean BMIs classified as obesity class I (30.0–34.9), and patients from DBG were also very close to this classification, presenting a mean BMI of 35.2. This significant reduction in BMI obtained through lifestyle changes in line with WHO guidelines is crucial to prevent premature death, cardiovascular and joint diseases, cancer, and diabetes ([37]).

The intervention was efficient in significantly reducing patients’ BMI. Patients with DEP, BED, and DEP + BED presented a similar reduction in BMI compared to the group without DEP and/or BED. Although it has been reported that the presence of mental disorders in patients with obesity may negatively impact weight loss, this usually happens due to the low motivation to treatment adherence ([17]). In this sense, an inpatient intervention presents as a good treatment option to ensure adherence ([28]).

The comprehensive intrahospital treatment for patients with obesity was also efficient in reducing all other anthropometric parameters evaluated in the 4 research groups. Waist circumference, in particular, is associated with increased cardiovascular risk at any given BMI ([5]) and is also more sensitive to lifestyle changes than BMI itself ([30]). The positive result of the intervention in WC highlights its potential to preserve the cardiovascular health of patients with obesity. The combination of individualized diet and exercise plans, together with personalized mental health support, was probably responsible for these results, since high levels of physical activity, frequent body weight monitoring, and adherence to a reduced-calorie diet are consistently associated with weight loss in individuals with obesity ([32]).

Waist circumference (WC) is a sensitive indicator of visceral adiposity and cardiometabolic risk, and it often responds more rapidly to lifestyle modification than BMI itself ([30]). However, in the present study, no significant correlation was found between the percentage change in WC and the duration of inpatient treatment. This finding suggests that improvements in abdominal fat distribution may occur independently of treatment length, potentially influenced by baseline metabolic status, sex-related fat distribution, or differential adherence to nutritional and physical activity components. Although increased WC has been linked to depressive symptoms and adverse mental health outcomes in individuals with obesity ([41]), our data indicate that the reduction in central adiposity observed during hospitalization was not directly associated with the duration of care, highlighting the multifactorial nature of body composition responses during inpatient interventions.

The large partial eta squared observed for the time effect (before and after the intervention) indicates that most of the variance in the dependent variables was explained by changes across measurement points. Such values are typical in within-subjects designs, where intra-individual variability is substantially smaller than between-subject variability, resulting in higher estimates of effect size. Therefore, the large η^2^_p_ primarily reflects the consistency of participants’ responses over time rather than an overestimation of the intervention effect.

Despite these findings, this study has some limitations. First, patients were not followed after discharge, so data on the maintenance of the newly acquired healthy habits is unknown. Secondly, the diagnoses of depression and binge eating disorder were established during a single admission interview, without standardized assessments, and focused on DEP and BED (not a full DSM-5 diagnostic screening). Information on medications and psychotherapy was not available for the group of patients, as they were individualized interventions. Third, data on body composition was unavailable. As in any weight management control program for obese patients, some lean mass must have been lost ([31]). However, to minimize this side effect, patients performed daily sessions of supervised resistance exercises as part of the treatment, besides following a diet program cautiously planned to provide them with the suitable amount of protein intake. Participants with obesity-related comorbidities such as type 2 diabetes and hypertension were included in the sample to maintain the clinical representativeness of the sample. However, because these conditions were not the primary focus and appeared similarly distributed across groups, no separate analyses were conducted.

## 5. Conclusions

In conclusion, patients with depression, binge-eating disorder, and comorbid DEP + BED achieved comparable and clinically meaningful reductions in BMI and waist circumference relative to the control group, with weight loss directly proportional to the length of inpatient treatment. These findings suggest that multidisciplinary inpatient programs can be equally effective for individuals with obesity and coexisting psychiatric conditions, highlighting the therapeutic potential of structured environments that integrate nutritional, physical, and psychological interventions. The association between treatment duration and body weight reduction underscores the importance of program adherence and continuity of care. Future studies should employ structured diagnostic interviews to comprehensively screen for psychiatric comorbidities and explore long-term outcomes after discharge to identify psychological or behavioral mediators that sustain weight management across different mental health profiles.

## Figures and Tables

**Figure 1 behavsci-15-01562-f001:**
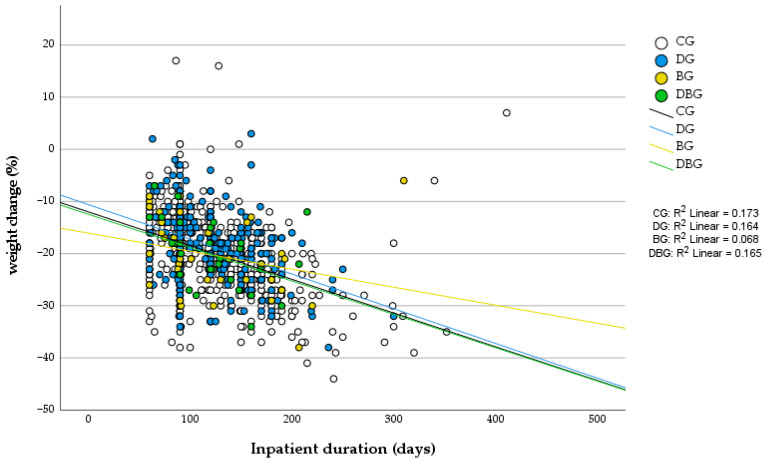
Scatter plot of inpatient duration (x-axis) versus percent weight change (y-axis) for each diagnostic group. DG = depression group, BG = binge-eating disorder group, DBG = depression and binge-eating disorder group, CG = group without DEP and/or BED.

**Figure 2 behavsci-15-01562-f002:**
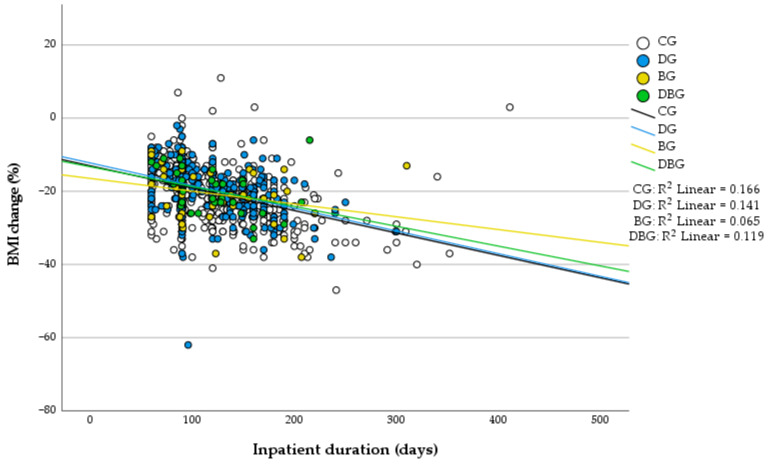
Scatter plot of inpatient duration (x-axis) versus percent BMI change (y-axis) for each diagnostic group. DG = depression group, BG = binge-eating disorder group, DBG = depression and binge-eating disorder group, CG = group without DEP and/or BED.

**Table 1 behavsci-15-01562-t001:** Characterization of demographic, lifestyle, and time of hospitalization data.

	DG (n = 279)	BG (n = 58)	DBG (n = 38)	CG (n = 622)	*p*
Gender (%F)	226 (81)	42 (72.4)	30 (78.9)	390 (62.7)	<0.001
Sedentary	245 (87.8)	48 (82.8)	31 (81.6)	509 (81.8)	NS
Age (years)	48.3 ± 16.3 ^a^	43.1 ± 15.5 ^abc^	39.4 ± 13.5 ^b^	46.9 ± 16.4 ^ac^	0.004
Hospitalization (days)	117.7 ± 43.4	118.5 ± 54.2	120.3 ± 40.0	118.5 ± 50.5	NS

F = females, DG = depression group, BG = binge-eating disorder group, DBG = depression and binge-eating disorder group, CG = without DEP and/or BED group, NS = non-significant difference. Data expressed as n (%) or means ± standard deviations. Different letters mean a statistically significant difference.

**Table 2 behavsci-15-01562-t002:** Comparison of body composition variables before and after the treatment.

	DG (n = 279)		BG (n = 58)		DBG (n = 38)		CG (n = 622)			
	Before	After	Before	After	Before	After	Before	After	*p*	η^2^_p_
BMI	42.9 ± 6.2	34.2 ± 4.2	44.6 ± 6.3	35.2 ± 5.6	42.3 ± 4.2	33.8 ± 3.9	43.3 ± 5.4	34.2 ± 4.7	<0.001 *	0.914
WC	115.2 ± 12.8	102.1 ± 12.0	117.5 ± 15.2	98.4 ± 9.1	113.9 ± 9.4	106.1 ± 8.3	116.7 ± 12.4	104.9 ± 10.2	<0.001 *	0.843
HC	131.5 ± 11.5	122.0 ± 9.9	134.7 ± 14.6	115.4 ± 10.2	130.3 ± 8.3	117.7 ± 8.0	131.9 ± 11.4	121.1 ± 8.5	<0.001 *	0.853
CC	43.3 ± 4.4	40.6 ± 4.4	44.4 ± 4.5	42.8 ± 4.1	44.0 ± 3.8	40.8 ± 3.8	44.4 ± 4.6	42.5 ± 4.7	<0.001 *	0.838

BMI = body mass index (kg/m^2^), WC = waist circumference (cm), HC = hip circumference (cm), CC = calf circumference (cm), DG = depression group, BG = binge-eating disorder group, DBG = depression + binge-eating disorder group, CG = control group without DEP and/or BED. η^2^_p_ = effect size for time (partial eta squared). Data expressed means ± standard deviations. * Comparison before and after the intervention, regardless of group.

**Table 3 behavsci-15-01562-t003:** Relative changes in weight, BMI, and WC after the intervention.

	DG (n = 279)	BG (n = 58)	DBG (n = 38)	CG (n = 622)	*p*
% weight change	−18.7 ± 6.9	−20.3 ± 7.2	−20.3 ± 6.3	−19.9 ± 7.6	0.131
% BMI change	−19.8 ± 6.9	−20.7 ± 7.4	−19.9 ± 6.3	−20.5 ± 7.4	0.447
% WC change	−10.9 ± 6.6	−9.4 ± 5.0	−11.0 ± 5.3	−12.1 ± 7.0	0.529

DG = depression group, BG = binge-eating disorder group, DBG = depression and binge-eating disorder group, CG = without DEP and/or BED group, % change = (after–before), BMI = body mass index, WC = waist circumference. Data expressed as means ± standard deviations.

**Table 4 behavsci-15-01562-t004:** Association between changes in weight, BMI, WC, and length of treatment.

	% Weight Change × TL		% BMI Change × TL		% WC Change × TL	
	r	*p*	r	*p*	r	*p*
DG	−0.404	<0.001	−0.375	<0.001	0.148	0.490
BG	−0.260	0.084	−0.254	0.092	−0.127	0.839
DBG	−0.407	0.023	−0.345	0.05	−0.113	0.852
CG	−0.416	<0.001	−0.416	<0.001	−0.128	0.499

DG = depression group, BG = binge-eating disorder group, DBG = depression and binge-eating disorder group, CG = group without DEP and/or BED, TL = treatment length, BMI = body mass index, WC = waist circumference, % change = (after–before).

## Data Availability

The dataset analyzed in this study consists of anonymized and aggregated patient data. Due to privacy and ethical restrictions, individual-level data cannot be publicly shared. Anonymized aggregate data may be made available from the corresponding author upon reasonable request and with permission from the institutional ethics committee.

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
