# Peer review of "Impact of Inpatient Treatment for Obesity in Patients with Comorbid Psychiatric Disorders"

_behavsci, 2025, doi:10.3390/bs15111562_

Round 1
Reviewer 1 Report
Comments and Suggestions for Authors
Manuscript Review
Overall, this manuscript examines the impact of an inpatient lifestyle-based intervention program for the treatment of obesity in persons with and without depression and binge eating. This study is of interest to clinicians and demonstrates how an inpatient program can reduce weight and other anthropometric measures, regardless of a depression or binge eating diagnosis.
However, there was concern with how depression and binge eating were only assessed during the diagnostic evaluation and not during or after completion of the lifestyle program. Given the authors’ premise that depression and binge eating can negatively impact weight, it would have been important to assess these symptoms, especially post-intervention as it may predict sustainment of weight loss and lifestyle habits.
To that end, additional recommendations are detailed below to strengthen the manuscript.
- Title: The title is a little misleading because the authors do not assess depression and binge eating symptoms during the intervention, so it is unclear how these symptoms influence the success of the program.
- Introduction: To strengthen the introduction, it is recommended the authors discuss more about the relationship between weight bias/weight stereotypes and depression and binge eating. Given that shame related to binge eating and weight may perpetuate binge eating and depression, it is important to draw more attention to this relationship.
- Methods: It is unclear if the 4 groups were divided only based on diagnosis. Other than diagnosis, did the groups differ? For example, did the depression group receive interventions that addressed depressed mood? Or did the binge eating group receive education on how their mood and binges may be related? How are these groups different from the control group?
- Methods: Were depression and binge eating symptoms assessed only at baseline? If so, was this only captured in the clinical interview with the psychologist or psychiatrist? Or were objective measures, such as symptom questionnaires given? It would be more compelling to see if changes occurred in these symptoms since the authors discuss how these symptoms can interfere with weight loss and sustainment of weight loss.
- Discussion: The authors describe an integrated approach to addressing psychiatric comorbidities. It would be helpful to clarify this further, as it seems unclear if the integrated approach was just the psychiatric assessment by a psychologist or psychiatrist or included other approaches.
Author Response
Comment 1: Overall, this manuscript examines the impact of an inpatient lifestyle-based intervention program for the treatment of obesity in persons with and without depression and binge eating. This study is of interest to clinicians and demonstrates how an inpatient program can reduce weight and other anthropometric measures, regardless of a depression or binge eating diagnosis.
However, there was concern with how depression and binge eating were only assessed during the diagnostic evaluation and not during or after completion of the lifestyle program. Given the authors’ premise that depression and binge eating can negatively impact weight, it would have been important to assess these symptoms, especially post-intervention as it may predict sustainment of weight loss and lifestyle habits.
Reply 1: Patients were evaluated by a licensed psychiatrist every month during treatment, including at the end of the program. However, the diagnosis did not change, and severity of symptoms were assessed by interview (no formal research instrument), as the aim of the present study was to evaluate the efficacy of the intervention in the management of obesity.
Comment 2: To that end, additional recommendations are detailed below to strengthen the manuscript. Title: The title is a little misleading because the authors do not assess depression and symptoms during the intervention, so it is unclear how these symptoms influence the success of the program.
Reply 2: The title has been adjusted. The presence of mental disorders in patients with obesity may worsen adherence to lifestyle interventions (e.g. Hoffmann et al., 2022), however, no study so far has investigated the impact of an inpatient intervention aimed at reducing body weight in persons with obesity who also struggle with depression and/or binge eating disorder.
Hoffmann, K., Kopciuch, D., Michalak, M., Bryl, W., Kus, K., Marzec, K., Raakow, J., Pross, M., Berghaus, R., Nowakowska, E., Kostrzewska, M., Zaprutko, T., Ratajczak, P., & Paczkowska, A. (2022). Adherence of obese patients from Poland and Germany and its impact on the effectiveness of morbid obesity treatment. Nutrients, 14(18), 3880.
Comment 3: Introduction: To strengthen the introduction, it is recommended the authors discuss more about the relationship between weight bias/weight stereotypes and depression and binge eating. Given that shame related to binge eating and weight may perpetuate binge eating and depression, it is important to draw more attention to this relationship.
Reply 3: We have included this discussion in the introduction, as suggested.
Comment 4: Methods: It is unclear if the 4 groups were divided only based on diagnosis. Other than diagnosis, did the groups differ? For example, did the depression group receive interventions that addressed depressed mood? Or did the binge eating group receive education on how their mood and binges may be related? How are these groups different from the control group?
Reply 4: Group division was were based on diagnosis. Control group did not present depression nor binge eating disorder. The program was the same for all the 4 groups: low-calorie diet and intensive lifestyle changes (daily physical activities, psychotherapy, educational interventions and individualized medical support). This information is highlighted in the manuscript.
Comment 5: Methods: Were depression and binge eating symptoms assessed only at baseline? If so, was this only captured in the clinical interview with the psychologist or psychiatrist? Or were objective measures, such as symptom questionnaires given? It would be more compelling to see if changes occurred in these symptoms since the authors discuss how these symptoms can interfere with weight loss and sustainment of weight loss.
Reply 5: Diagnosis of depression and binge-eating disorder was confirmed by a physician at admission, through clinical interview according to DSM-5 (APA, 2013), and patients underwent monthly individual appointments with this professional. Nonetheless, we do not have data on symptoms change as they were reassessed by an interview, not by a formal research instrument.
American Psychiatric Association. (2013). Diagnostic and statistical manual of mental disorders (5th ed.). American Psychiatric Publishing.
Comment 6: Discussion: The authors describe an integrated approach to addressing psychiatric comorbidities. It would be helpful to clarify this further, as it seems unclear if the integrated approach was just the psychiatric assessment by a psychologist or psychiatrist or included other approaches.
Reply 6: The integrated approach refers to the comprehensive intervention to treat patients with obesity and psychiatric comorbidities. It involved a multidisciplinary team: psychiatrists, general practitioners, psychologists, nurses, nutritionists, physical therapists, and physical educators.
Reviewer 2 Report
Comments and Suggestions for Authors
Dear Authors,
Thanks of this study, which aims to evaluate the effectiveness of inpatient treatment for obesity in people with and without depression and binge-eating disorder. However, the manuscript requires substantial revisions.
- The current title of the manuscript does not accurately reflect the objectives or findings of the study. Specifically, the study focuses on the impact of inpatient treatment on obesity in four groups: control, depression, binge-eating disorder and comorbid depression with binge-eating disorder. However, the existing title suggests that the primary focus is on evaluating the influence of depression and binge-eating disorder on the outcomes of inpatient treatment. To ensure consistency and clarity, the title should be revised to better reflect the study’s purpose and results.
- The introduction does not provide a sufficient overview of previous research conducted in Brazil on this topic. In particular, it lacks a synthesis of studies addressing eating disorders, body mass index (BMI) outcomes and depression scores in the Brazilian context. Incorporating this literature would reinforce the rationale behind the study and position the research within the existing national body of evidence.
- The materials and methods section of the manuscript lacks clarity and requires substantial revision. Several aspects remain ambiguous or are not described well enough:
- The term 'low-calorie diet' must be explicitly defined. The daily caloric intake, macronutrient distribution and dietary guidelines applied should be specified.
- Eating behaviour: Please clarify the number, timing and type of meals provided or recommended.
- Sleep patterns: Indicate the parameters (e.g. duration, quality and consistency).
- Physical activity: Provide details of the types of activities, their duration, frequency and intensity.
- Specify whether the intervention included pharmacological treatment, and if so, which medications were administered.
- Clarify whether all groups were subject to the same intervention protocols. In particular, indicate whether patients with depression received psychiatric consultations or additional support.
- State whether participants with other medical conditions (e.g., diabetes) were included, and how these conditions were accounted for in the analysis.
- Clearly outline the inclusion and exclusion criteria applied to select participants.
- The discussion section fails to address the reasons underlying the favourable outcomes demonstrated by all patient groups. Although factual results are presented, the absence of interpretative analysis or possible explanatory mechanisms means the discussion lacks depth and critical engagement.
- The conclusion section needs to be improved as it just repeats facts without offering any deeper analysis, critical interpretation or consideration of the wider implications of the findings.
Author Response
Comment 1: Thanks of this study, which aims to evaluate the effectiveness of inpatient treatment for obesity in people with and without depression and binge-eating disorder. However, the manuscript requires substantial revisions.
The current title of the manuscript does not accurately reflect the objectives or findings of the study. Specifically, the study focuses on the impact of inpatient treatment on obesity in four groups: control, depression, binge-eating disorder and comorbid depression with binge-eating disorder. However, the existing title suggests that the primary focus is on evaluating the influence of depression and binge-eating disorder on the outcomes of inpatient treatment. To ensure consistency and clarity, the title should be revised to better reflect the study’s purpose and results.
Reply 1: The title has been changed according to the suggestion.
Comment 2: The introduction does not provide a sufficient overview of previous research conducted in Brazil on this topic. In particular, it lacks a synthesis of studies addressing eating disorders, body mass index (BMI) outcomes and depression scores in the Brazilian context. Incorporating this literature would reinforce the rationale behind the study and position the research within the existing national body of evidence.
Reply 2: The overview of national literature has been added as suggested and is highlighted in the manuscript.
Comment 3: The materials and methods section of the manuscript lacks clarity and requires substantial revision. Several aspects remain ambiguous or are not described well enough:
The term 'low-calorie diet' must be explicitly defined. The daily caloric intake, macronutrient distribution and dietary guidelines applied should be specified.
Reply 3: Caloric intake and macronutrient distribution were individualized, following the Brazilian guidelines for the management of obesity. This information is highlighted in the manuscript.
Comment 4: Eating behaviour: Please clarify the number, timing and type of meals provided or recommended.
Reply 4: Meal plans were individualized and changed along the course of the intervention, following the Brazilian guidelines for the management of obesity.
Comment 5: Sleep patterns: Indicate the parameters (e.g. duration, quality and consistency).
Reply 5: As patients presented varied sleep patterns at baseline, plans were individualized, targeting good habits of sleep hygiene and at least 8 hours of sleep per night.
Comment 6: Physical activity: Provide details of the types of activities, their duration, frequency and intensity.
Reply 6: Physical activity was individualized. Its duration, frequency and intensity changed along the course of the intervention, following the Brazilian guidelines for the management of obesity. Types of activities included weight training, water aerobics and walking. The target was a minimum of 50 minutes of moderate-intensity exercise per day.
Comment 7: Specify whether the intervention included pharmacological treatment, and if so, which medications were administered.
Reply 7: The intervention did not include pharmacological treatment for obesity. Nonetheless, patients kept their usual medications for other conditions (e.g. gastroesophageal reflux, musculoskeletal pain, etc.).
Comment 8: Clarify whether all groups were subject to the same intervention protocols. In particular, indicate whether patients with depression received psychiatric consultations or additional support.
Reply 8: All patients (including the ones in control group) underwent the same intervention protocol and monthly individual appointments with a psychiatrist and weekly sessions with a psychologist.
Comment 9: State whether participants with other medical conditions (e.g., diabetes) were included, and how these conditions were accounted for in the analysis.
Reply 9: Participants with obesity-related comorbidities (e.g., type 2 diabetes, hypertension, dyslipidemia) were included to maintain the clinical representativeness of the inpatient population. Because these conditions were not the primary focus and were similarly distributed among groups, no separate analyses were conducted. This limitation has been acknowledged in the revised manuscript.
Comment 10: Clearly outline the inclusion and exclusion criteria applied to select participants.
Reply 10: Inclusion and exclusion criteria were added and are highlighted in the methods section.
Comment 11: The discussion section fails to address the reasons underlying the favourable outcomes demonstrated by all patient groups. Although factual results are presented, the absence of interpretative analysis or possible explanatory mechanisms means the discussion lacks depth and critical engagement.
Reply 11: Possible explanatory mechanisms on the results of the intervention are highlighted in the discussion section, as suggested.
Comment 12: The conclusion section needs to be improved as it just repeats facts without offering any deeper analysis, critical interpretation or consideration of the wider implications of the findings.
Reply 12: The conclusion has been changed to offer a deeper analysis, critical interpretation and implications of the findings.
Reviewer 3 Report
Comments and Suggestions for Authors
This manuscript addresses an important and timely topic, namely the impact of depression and binge eating disorder on inpatient treatment outcomes in obesity. The study leverages a large clinical sample and examines a clinically meaningful question, making the work potentially valuable to the field. However, several methodological and reporting issues substantially limit the interpretability of the current findings. In particular, key aspects of diagnostic procedures, program description, statistical analyses, and reporting clarity require further elaboration and refinement. Without these revisions, it is difficult to draw firm conclusions regarding the validity and generalizability of the results. I provide detailed comments below with the aim of strengthening the methodological rigor and clarity of presentation.
1. Definition of the DEP Group
- The manuscript states that depression was diagnosed via a DSM-5-based clinical interview. Using the broader term “Depressive disorders” is acceptable only if subcategories are clearly reported. DSM-5 includes Major Depressive Disorder, Persistent Depressive Disorder, Disruptive Mood Dysregulation Disorder, and Premenstrual Dysphoric Disorder. The authors should specify which DSM-5 categories were diagnosed and the number of participants in each category.
2. Psychiatric Diagnoses
- Were all psychiatric assessments conducted by licensed psychiatrists?
- For the 997 participants, the authors should report results for disorders other than depression and BED. For example, DSM-5 interviews are expected to assess other eating disorders (e.g., bulimia nervosa), bipolar disorder, substance-use disorders, etc.
- Clarify whether participants in the control group included individuals with other psychiatric conditions. Specify the number and type of such cases.
- Indicate whether including or excluding these participants with other psychiatric disorders affects the study results.
3. Psychological Assessments
- If available, include standardized scales for depression and binge eating symptoms (e.g., PHQ-9, BDI, BES, EDE).
- These data can be reported in Table 1. If not available, any estimates of symptom severity may be mentioned.
4. Medication and Psychotherapy
- Report whether participants with depression or other psychiatric disorders received pharmacological treatment. Specify the number of participants and drug types.
- Provide details on psychotherapy tailored to patients with depression and/or BED, if applicable.
- Was there any clinical outcome related with psychiatric conditions (e.g., depression, binge eating symptoms, etc.)?
5. Inclusion and Exclusion Criteria
- Both Criteria should be described more explicitly.
- How many participants were excluded due to treatment duration <1 month? Was the dropout rate consistent across DG, BG, DBG, and CG groups? Participants with depression or BED may have lower adherence. The manuscript should clarify whether the treatment effects are interpretable only among completers.
6. Inpatient Program Description
-The inpatient program in this study is very effective and noteworthy. Detailed program description (diet, exercise, education, daily schedule, etc.) would be highly valuable for replication. Please include the degree of participant autonomy in leisure, meals, and activities, as such factors could influence outcomes across groups.
7. Discharge Criteria
- Specify discharge conditions. Were participants discharged solely at their own discretion, or were specific weight loss targets set and monitored by clinical staff?
8. Adherence
- Provide additional adherence data or individual variation during the inpatient period.
9. Sedentary Behavior
- Define sedentary behavior explicitly and mention it in Table 1 as well (Was it defined as "not achieving 150 minutes per week of physical activity"?)
- Did sedentary behavior change during hospitalization, or was a non-sedentary lifestyle maintained across participants?
10. Statistical Analysis and Results
- It may be helpful to consult a statistician to ensure the appropriateness of the methods.
- Please provide effect sizes together with p-values, if available.
- The Methods section states that anthropometric comparisons were conducted using two-way ANOVA. Since measurements are repeated within subjects, a two-way mixed-design ANOVA is more appropriate. Alternatively, one
- Group differences in age and gender necessitate controlling these variables in subsequent analyses. ANCOVA with age and gender as covariates is recommended. Separate analyses could examine whether these factors influence treatment outcomes.
- Table 2: The p-values should clarify whether they reflect the main effect of time (before vs. after). In addition, group differences at baseline should be examined. Consider using the mixed-design ANOVA suggested above. Alternatively, analyze baseline differences with one-way ANOVA and post-intervention change scores with one-way ANOVA (as in current Table 3).
- Table 3: Relative weight change (% change) is more informative than absolute change. Include waist circumference, as it is relevant for metabolic outcomes.
- Table 4: Similarly, include waist changes. Scatter plots of inpatient duration (x-axis) vs. weight change (y-axis) with separate markers and trend lines for each group would clarify whether weight loss rates differ among groups.
11. Discussion
- Please expand discussion of prior studies. Critically compare and contrast findings with existing literature, noting both consistencies and discrepancies. Discuss the study’s significance, applications, and future research directions.
- Extend the Conclusion to highlight the study’s importance and broader implications.
12. References
- Please check the cited references carefully.
- For example, DSM-5 (2013) and DSM-5-TR (2022, Text Revision) are distinct editions and should not share the same title/edition; label DSM-5-TR explicitly as “Text Revision” with the correct year and publisher details. If there is no specific methodological rationale, use a single edition consistently across the manuscript and harmonize all in-text citations and reference entries accordingly.
- Please also conduct a brief reference audit to identify and fix any remaining inaccuracies (e.g., mismatched years, duplicated titles, missing DOIs, inconsistent journal names/access dates).
1. Paragraph Structure
- Reorganize paragraphs to match topic and logical flow.
- Avoid redundancy. For example, two consecutive paragraphs repeat the same points about waist circumference being clinically salient and more responsive than BMI; these should be merged and streamlined to avoid duplication and improve flow.
- Single-sentence paragraphs should be merged with related content for clarity.
2. Clarity and Consistency of Expression
- Make claims precise and aligned with analyses (e.g., replace “equally significant improvements” with “no between-group differences in change were detected,” pending a group×time interaction test in a mixed design).
- Strengthen integration of tables/figures into the narrative by explicitly linking results to interpretive statements.
- Please check the consistency and coherence across the whole manuscript thoroughly. For example, the Methods define Pearson correlation strength categories (e.g., weak, moderate, strong, very strong), but these classifications are not explicitly applied in the Results and Discussion. Either remove this description from the Methods, or ensure that the same classification scheme is consistently applied in the subsequent sections. In addition, please pay close attention to consistent use of terminology and abbreviations throughout the manuscript (e.g., group labels, clinical terms, and statistical notation).
Author Response
Comment 1: The manuscript states that depression was diagnosed via a DSM-5-based clinical interview. Using the broader term “Depressive disorders” is acceptable only if subcategories are clearly reported. DSM-5 includes Major Depressive Disorder, Persistent Depressive Disorder, Disruptive Mood Dysregulation Disorder, and Premenstrual Dysphoric Disorder. The authors should specify which DSM-5 categories were diagnosed and the number of participants in each category.
Reply 1: All categories of DSM-5 were included as a single group, and unfortunately we did not keep a detailed record, which prevents the disclosure of the number of patients in each one.
Comment 2: Psychiatric Diagnoses
- Were all psychiatric assessments conducted by licensed psychiatrists?
Reply 2: Yes, they were.
Comment 3: For the 997 participants, the authors should report results for disorders other than depression and BED. For example, DSM-5 interviews are expected to assess other eating disorders (e.g., bulimia nervosa), bipolar disorder, substance-use disorders, etc.
Reply 3: All categories of DSM-5 were included as a single group, and unfortunately we did not keep a detailed record, which prevents the disclosure of the number of patients in each one.
Comment 4: Clarify whether participants in the control group included individuals with other psychiatric conditions. Specify the number and type of such cases.
Reply 4: Participants in the control group did not present any psychiatric conditions. This was included and highlighted in the manuscript.
Comment 5: Indicate whether including or excluding these participants with other psychiatric disorders affects the study results.
Reply 5: Participants in the control group did not present any psychiatric conditions.
Comment 6: Psychological Assessments
- If available, include standardized scales for depression and binge eating symptoms (e.g., PHQ-9, BDI, BES, EDE).
- These data can be reported in Table 1. If not available, any estimates of symptom severity may be mentioned.
Reply 6: Symptoms severity was assessed by interview along the study course. Unfortunetely, we did not employ any standardized scales.
Comment 7: Medication and Psychotherapy
- Report whether participants with depression or other psychiatric disorders received pharmacological treatment. Specify the number of participants and drug types.
Reply 7: Participants reached the hospital to treat obesity, and the ones who had been prescribed pharmacological treatment kept their usual medication. Despite the psychiatric and psychological support offered to all patients, we do not have a record about the drugs used by patients, as the focus of the treatment was in obesity.
Comment 8: Provide details on psychotherapy tailored to patients with depression and/or BED, if applicable.
Reply 8: Psychotherapy was offered to patients in individual weekly sessions, and was individualized according to each participant’s needs.
Comment 9: Was there any clinical outcome related with psychiatric conditions (e.g., depression, binge eating symptoms, etc.)?
Reply 9: No, as psychiatric outcomes were only assessed by interview with a licensed psychiatrist.
Comment 10:
Inclusion and Exclusion Criteria
- Both Criteria should be described more explicitly.
Reply 10: They were added and highlighted in the methods section.
Comment 11: How many participants were excluded due to treatment duration <1 month? Was the dropout rate consistent across DG, BG, DBG, and CG groups? Participants with depression or BED may have lower adherence. The manuscript should clarify whether the treatment effects are interpretable only among completers.
Reply 11: Nine patients were excluded due to treatment duration lower than 1 month. 3 of them belonged to CG, 2 presented DEP, 3 DEP+BED and 3 patients had BED diagnosis. As dropout was very low, similar among groups and mainly motivated by financial reasons (not related to the treatment), we did not present this data in the manuscript.
Comment 12:
Inpatient Program Description
-The inpatient program in this study is very effective and noteworthy. Detailed program description (diet, exercise, education, daily schedule, etc.) would be highly valuable for replication. Please include the degree of participant autonomy in leisure, meals, and activities, as such factors could influence outcomes across groups.
Reply 12: The program was individualized, and patients had personal goals and prescription for diet, physical activities, etc. They received an agenda of daily activities, but they could change the schedule of the physical activities sessions if they preferred. On weekends, they were offered therapeutic leisure activities in which they could participate or not. This flexibility was not offered for meals.
Comment 13: Discharge Criteria
- Specify discharge conditions. Were participants discharged solely at their own discretion, or were specific weight loss targets set and monitored by clinical staff?
Reply 13: Discharge occurred after reaching the target BMI or time of treatment set at admission.
Comment 14: Adherence
- Provide additional adherence data or individual variation during the inpatient period.
Reply 14: As they were all inpatients, there was no variation in adherence. All 997 patients completed the program.
Comment 15: Sedentary Behavior
- Define sedentary behavior explicitly and mention it in Table 1 as well (Was it defined as "not achieving 150 minutes per week of physical activity"?)
Reply 15: Yes, it was. This was added to the text.
Comment 16: Did sedentary behavior change during hospitalization, or was a non-sedentary lifestyle maintained across participants?
Reply 16: Yes, it did. As the program included daily sessions of physical activity all patients became active individuals.
Comment 17. Statistical Analysis and Results
- It may be helpful to consult a statistician to ensure the appropriateness of the methods.
- Please provide effect sizes together with p-values, if available.
Reply 17: We had a statistician revise the analysis, add the effect sizes (highlighted in table 2), % changes in weight, BMI and WC, and scatter plots.
Comment 18: The Methods section states that anthropometric comparisons were conducted using two-way ANOVA. Since measurements are repeated within subjects, a two-way mixed-design ANOVA is more appropriate. Alternatively, one
Reply 18: The comment appears to have reached us incomplete, but we have used 2-way ANOVA for repeated measures, this was clarified in the text.
Comment 19: Group differences in age and gender necessitate controlling these variables in subsequent analyses. ANCOVA with age and gender as covariates is recommended. Separate analyses could examine whether these factors influence treatment outcomes.
Reply 19: We conducted ANCOVA including age and gender as covariates to control for baseline demographic differences across groups. The results showed that neither age nor gender significantly influenced BMI, and that the main effect of Group was not significant after adjustment (p=0.670). These findings confirm that the previously observed differences were not driven by demographic factors.
Comment 20: Table 2: The p-values should clarify whether they reflect the main effect of time (before vs. after). In addition, group differences at baseline should be examined. Consider using the mixed-design ANOVA suggested above. Alternatively, analyze baseline differences with one-way ANOVA and post-intervention change scores with one-way ANOVA (as in current Table 3).
Reply 20: P-values refer to the effect of time, this was highlighted on table 2.
Comment 21: Table 3: Relative weight change (% change) is more informative than absolute change. Include waist circumference, as it is relevant for metabolic outcomes.
Reply 21: Data in table 3 has been changed accordingly.
Comment 22: Table 4: Similarly, include waist changes. Scatter plots of inpatient duration (x-axis) vs. weight change (y-axis) with separate markers and trend lines for each group would clarify whether weight loss rates differ among groups.
Reply 22: Data on waist circumference has been added to table 4. Scatter plots (figures 1 and 2) were also added to the manuscript.
Comment 23: Discussion
- Please expand discussion of prior studies. Critically compare and contrast findings with existing literature, noting both consistencies and discrepancies. Discuss the study’s significance, applications, and future research directions.
Reply 23: Discussion has been expanded according to the suggestion.
Comment 24: Extend the Conclusion to highlight the study’s importance and broader implications.
Reply 24: The conclusion has been changed highlight the study’s importance and implications of the findings.
Comment 25: References
- Please check the cited references carefully.
Reply 25: We have revised the reference list.
Comment 26: For example, DSM-5 (2013) and DSM-5-TR (2022, Text Revision) are distinct editions and should not share the same title/edition; label DSM-5-TR explicitly as “Text Revision” with the correct year and publisher details. If there is no specific methodological rationale, use a single edition consistently across the manuscript and harmonize all in-text citations and reference entries accordingly.
Reply 26: We appreciated the thorough observation. The correction has been made.
Comment 27: Please also conduct a brief reference audit to identify and fix any remaining inaccuracies (e.g., mismatched years, duplicated titles, missing DOIs, inconsistent journal names/access dates).
Reply 27: We have revised the reference list.
Comment 28: Comments on the Quality of English Language
Paragraph Structure
- Reorganize paragraphs to match topic and logical flow.
Reply 28: The manuscript has been revised accordingly.
Comment 29: Avoid redundancy. For example, two consecutive paragraphs repeat the same points about waist circumference being clinically salient and more responsive than BMI; these should be merged and streamlined to avoid duplication and improve flow.
Reply 29: This part of the text has been revised.
Comment 30: Single-sentence paragraphs should be merged with related content for clarity.
Reply 30: The manuscript has been revised accordingly.
Comment 31: Clarity and Consistency of Expression
- Make claims precise and aligned with analyses (e.g., replace “equally significant improvements” with “no between-group differences in change were detected,” pending a group×time interaction test in a mixed design).
Reply 31: This part of the text has been re-written and is highlighted in the manuscript.
Comment 32: Strengthen integration of tables/figures into the narrative by explicitly linking results to interpretive statements.
Reply 32: We have revised the text in the results section to match this recommendation.
Comment 33: Please check the consistency and coherence across the whole manuscript thoroughly. For example, the Methods define Pearson correlation strength categories (e.g., weak, moderate, strong, very strong), but these classifications are not explicitly applied in the Results and Discussion. Either remove this description from the Methods or ensure that the same classification scheme is consistently applied in the subsequent sections. In addition, please pay close attention to consistent use of terminology and abbreviations throughout the manuscript (e.g., group labels, clinical terms, and statistical notation).
Reply 33: We have removed the strength categories classification of Pearson’s correlation, as suggested. The manuscript has been revised for the consistent use of terminology and abbreviations.
Round 2
Reviewer 1 Report
Comments and Suggestions for Authors
Thank you for addressing the reviewers' comments. It is a much stronger paper!
Author Response
We sincerely thank the reviewer for the positive feedback and for the insightful comments that helped us strengthen the manuscript. We are very pleased that the revised version meets the reviewer’s expectations.
Reviewer 2 Report
Comments and Suggestions for Authors
Dear Authors,
Thank you for providing a revised version of the manuscript and for taking the suggestions into account. The manuscript is now much improved, presenting the study in a clearer and more comprehensible manner.
Author Response

(The authors gave the same response as above.)

Reviewer 3 Report
Comments and Suggestions for Authors
I appreciate the authors’ efforts to address the previous comments and to substantially revise the manuscript. The responses indicate a sincere attempt to improve methodological transparency and reporting clarity. Nevertheless, several critical issues remain unresolved, and in some cases, the revisions.
(Major points)
1. Details for Methods.
- Please divide 2. Materials and Methods into subsections (e.g., 2.1. Participants, 2.2. Therapeutic interventions, 2.3. Measurement tools, 2.4. Statistical analysis, etc.) and provide more details on each. More details for the inpatient program should be provided as well for the replicability.
2. Psychiatric diagnosis
- The authors reponded that the control group (N=622) did not present any psychiatric conditions but it is very unlikely. Please clarify whether psychiatrists assessed all psychiatric diseases in DSM-5 in the participants with clinical interview or examined only depressive disorders and binge eating disorder in the particpiants. In the latter case, we cannot exclude the possiblity of other comorbid mental disorders. It should be precisely described in the manuscript in details (mentioned in limitations in Discussion section as well).
3. Data reliability
- In Figures 1 and 2, a couple of participants appear to show more than a 75% reduction in weight and BMI. Such extreme values are highly unusual and raise concerns about potential data processing errors. These anomalies may substantially affect the reliability and interpretability of the overall results. I recommend that the authors carefully verify these data points, provide clarification on possible causes (e.g., data recording or calculation errors), and provide raw data sufficient for data verification.
4. ANCOVA
- As four groups differ in gender ratio and age, ANOVA (with gender and age as covariate) should be used for all analysis. The current mansucript appears to use ANCOVA only in the initial comparison of BMI across 4 groups only.
(Minor points)
* Limitations: Following points should be addressed as the limitations of the study.
- Lack of standardized psychological assessments (e.g., PHQ-9, BDI, BES, or EDE-Q)
- Lack of information on psychotherapy and medications.
* The interpretation of the results appears to be only partially supported by statistical evidence and tends to be overstated.
- For example, a lower R² in the BED group does not necessarily indicate a weaker effect but rather greater individual variability. The authors should statistically compare the slopes (coefficient b) across groups to draw valid conclusions.
Author Response
(Major points)
Comment 1 - Details for Methods.
- Please divide 2. Materials and Methods into subsections (e.g., 2.1. Participants, 2.2. Therapeutic interventions, 2.3. Measurement tools, 2.4. Statistical analysis, etc.) and provide more details on each. More details for the inpatient program should be provided as well for the replicability.
Reply 1 - Section 2 has been divided into subsections and detailed according to the suggestion.
Comment 2 - Psychiatric diagnosis
- The authors responded that the control group (N=622) did not present any psychiatric conditions but it is very unlikely. Please clarify whether psychiatrists assessed all psychiatric diseases in DSM-5 in the participants with clinical interview or examined only depressive disorders and binge eating disorder in the participants. In the latter case, we cannot exclude the possibility of other comorbid mental disorders. It should be precisely described in the manuscript in details (mentioned in limitations in Discussion section as well).
Reply 2 - At admission, board-certified psychiatrists conducted clinical interviews focused on diagnosing DEP disorders and BED according to DSM-5 criteria. A comprehensive screening for all DSM-5 disorders was not performed; therefore, the control group should be interpreted as participants without current DEP and without BED at admission, rather than free of any psychiatric condition. We have clarified this in the Methods and acknowledged the implication in the Limitations.
Comment 3 - Data reliability
- In Figures 1 and 2, a couple of participants appear to show more than a 75% reduction in weight and BMI. Such extreme values are highly unusual and raise concerns about potential data processing errors. These anomalies may substantially affect the reliability and interpretability of the overall results. I recommend that the authors carefully verify these data points, provide clarification on possible causes (e.g., data recording or calculation errors), and provide raw data sufficient for data verification.
Reply 3 - After re-examining the raw data, we confirmed that these values correspond to participants who experienced substantial reductions in weight during extended hospitalizations (>180 days). Their records were cross-checked with clinical charts and no data-entry or processing errors were identified. Because these cases represented true clinical observations and not inconsistencies, they were retained in the analyses.
Comment 4 - ANCOVA
- As four groups differ in gender ratio and age, ANOVA (with gender and age as covariate) should be used for all analysis. The current manuscript appears to use ANCOVA only in the initial comparison of BMI across 4 groups only.
Reply 4 - We performed additional ANCOVA analyses including age and gender as covariates to examine their potential influence on the main anthropometric outcomes (weight and BMI). These analyses indicated that neither age (p = 0.246) nor gender (p = 0.271) were significant predictors, and that the main effect of group remained non-significant after adjustment (F(3,902) = 0.575, p = 0.670, η²â‚š = 0.365). Thus, age and gender did not meaningfully affect the results. Given their lack of influence, we retained the original ANOVA models for clarity and interpretability. This clarification has been added to the Statistical analysis section.
(Minor points)
Comment 5 - Limitations: Following points should be addressed as the limitations of the study.
- Lack of standardized psychological assessments (e.g., PHQ-9, BDI, BES, or EDE-Q)
- Lack of information on psychotherapy and medications.
Reply 5 - They were added as limitations and are highlighted in the manuscript.
Comment 6 - The interpretation of the results appears to be only partially supported by statistical evidence and tends to be overstated.
- For example, a lower R² in the BED group does not necessarily indicate a weaker effect but rather greater individual variability. The authors should statistically compare the slopes (coefficient b) across groups to draw valid conclusions.
Reply 6 - We agree that a lower R² value in the binge-eating disorder group reflects greater interindividual variability rather than necessarily indicating a weaker effect. To address this point, the description of the results and their interpretation were revised to clarify that the observed differences in R² represent differences in variance rather than in effect magnitude. Given that the slopes (b coefficients) of the regression lines were visually similar and their 95% confidence intervals overlapped substantially across groups, we refrained from asserting between-group differences in slope. This clarification has been incorporated into the Results and Discussion sections to ensure a more conservative interpretation consistent with the statistical evidence.
Round 3
Reviewer 3 Report
Comments and Suggestions for Authors
I appreciate the authors' effort in revision that significantly improved the scientific quality of the manuscript. Nevertheless, my concern regarding data reliability (Comment 3) remains unresolved; please refer to the prior communication and the authors’ Reply 3.
Comment 3 - Data reliability
- In Figures 1 and 2, a couple of participants appear to show more than a 75% reduction in weight and BMI. Such extreme values are highly unusual and raise concerns about potential data processing errors. These anomalies may substantially affect the reliability and interpretability of the overall results. I recommend that the authors carefully verify these data points, provide clarification on possible causes (e.g., data recording or calculation errors), and provide raw data sufficient for data verification.
Reply 3 - After re-examining the raw data, we confirmed that these values correspond to participants who experienced substantial reductions in weight during extended hospitalizations (>180 days). Their records were cross-checked with clinical charts and no data-entry or processing errors were identified. Because these cases represented true clinical observations and not inconsistencies, they were retained in the analyses.
There are two participants who showed >90% reduction in BMI in Figure 2 in the revised manuscript. For example, if the patients were 400 kg in baseline, they should be about 40 kg after weight reduction. Such magnitudes are extraordinary and raise justified concerns about possible data-processing or recording errors.
Please provide baseline and follow-up weight measurements for these two participants, and indicate measurement dates and units. I also recommend a thorough review for potential errors.
Author Response
Comment 1: I appreciate the authors' effort in revision that significantly improved the scientific quality of the manuscript. Nevertheless, my concern regarding data reliability (Comment 3) remains unresolved; please refer to the prior communication and the authors’ Reply 3. There are two participants who showed >90% reduction in BMI in Figure 2 in the revised manuscript. For example, if the patients were 400 kg in baseline, they should be about 40 kg after weight reduction. Such magnitudes are extraordinary and raise justified concerns about possible data-processing or recording errors. Please provide baseline and follow-up weight measurements for these two participants, and indicate measurement dates and units. I also recommend a thorough review for potential errors.
Reply 1: We sincerely thank the reviewer for carefully re-examining the figures and for pointing out this important issue. Upon a renewed inspection of the raw dataset, we identified a data-entry error in the final weight values of two participants from the control group, which resulted in the implausibly large reductions observed in Figure 2.
After reviewing the original clinical records, we confirmed that these two cases were due to typing errors in the discharge weights (decimal misplacement). The data were corrected, and all subsequent analyses involving body weight, BMI, and percentage change in these variables (Tables 2-4 and Figures 1-2) were recomputed accordingly and are highlighted in the manuscript.
The corrected dataset shows no extreme or biologically implausible values. Importantly, the overall results and interpretation of the study remain unchanged, as the corrected values did not materially affect the significance or direction of the findings.
We appreciate the reviewer’s attentive observation, which helped us improve the accuracy and reliability of our analyses.